# SARS-CoV-2 Surveillance in Hospital Wastewater: CLEIA vs. RT-qPCR

**Supranee Thongpradit** [1] , **Suwannee Chanprasertyothin** [1], **Ekawat Pasomsub** [2],
**Boonsong Ongphiphadhanakul** [1,3] **and Somsak Prasongtanakij** [1,*]

1   Research Center, Faculty of Medicine Ramathibodi Hospital, Mahidol University, Bangkok 10400, Thailand
2   Department of Pathology, Faculty of Medicine Ramathibodi Hospital, Mahidol University,
    Bangkok 10400, Thailand
3   Department of Medicine, Faculty of Medicine Ramathibodi Hospital, Mahidol University,
    Bangkok 10400, Thailand
*   Correspondence: somsak.pra@mahidol.edu; Tel.: +66-9557-09888

**Abstract:** The utilization of wastewater as a community surveillance method grew during the COVID-19 epidemic. COVID-19 hospitalizations are closely connected with wastewater viral signals, and increases in wastewater viral signals can serve as an early warning indication for rising hospital admissions. While reverse transcriptase quantitative polymerase chain reaction (RT-qPCR) is the most often used approach for detecting SARS-CoV-2 in wastewater, chemiluminescence enzyme immunoassay (CLEIA) is an alternative automated method. In two assays, 92 wastewater grab samples from a hospital were investigated for the presence of SARS-CoV-2, expected for continuous and monitoring SARS-CoV-2 surveillance. One was in the RT-qPCR nucleic acid test, and another was in the CLEIA assay quantitative antigen test. In 24/92 (26.09%) of the wastewater samples, RT-qPCR identified at least two SARS-CoV-2 genes (ORF1ab, N, or S genes). CLEIA, on the other hand, detected SARS-CoV-2 antigen in 39/92 (42.39%) of the samples. CLEIA demonstrated a low sensitivity and specificity of sensitivity of 54.2% (95% CI: 44.0–64.3%) and 61.8% (95% CI: 51.8–71.7%), respectively, as compared to RT-qPCR. The κ coefficient indicated slight agreement between assay. Then, the CLEIA assay cannot replace molecular-based testing like RT PCR for determining SARS-CoV-2 in hospital wastewater.

**Keywords:** COVID-19; SARS-CoV-2; hospital wastewater; CLEIA; RT-qPCR





## 1. Introduction

The severe acute respiratory syndrome coronavirus, also known as SARS-CoV, caused a sudden epidemic outbreak of coronavirus illness in 2019 (COVID-19), which has raised significant concerns among the general public, scientific community, and healthcare professionals worldwide, and has had a major detrimental effect on people's health, the economy, and society. COVID-19 hospitalizations are strongly linked to wastewater viral signals, and increases in wastewater viral signals can serve as an early warning indicator for growing hospital admissions [1,2]. Several papers have revealed the presence of SARS-CoV-2 RNA in stools from COVID-19 patients, as well as the presence of SARS-CoV-2 in wastewaters around the world [3–6]. Another avenue for COVID-19 transfer into water and wastewater is through the widespread usage of face masks by the general population, patients, and health personnel worldwide. Following their use, those face masks were discarded without treatment or disinfection, raising worries about potential health risks and affecting the environment [7]. A transmission channel through the sanitary (or wastewater) plumbing system may be responsible for the spread of COVID-19 within communities and for environmental contamination. Reports of SARS-CoV-2 in wastewater and water have been established [8,9]. In general, monitoring viral infections in wastewater presents various obstacles. The varied composition of wastewater matrices, the low concentration nature of biomarkers in wastewater, the difficulty of obtaining good sample locations, and the requirement for

effective virus-concentrating technologies often restrict this method's ability to produce quantitative predictions using viral RNA [10,11]. Despite these obstacles, several studies have detected SARS-CoV-2 in the feces of COVID-19 patients and in wastewater [3,5,12]. Furthermore, if high temporal resolution is followed by timely analysis and reporting, SARS-CoV-2 wastewater surveillance has the potential to serve as an early warning system for disease outbreaks in certain localities [13,14]. It is a non-invasive, community-wide surveillance technology that decreases selection bias by detecting subclinical illnesses [15]. Thus, the United States Centers for Disease Control and Prevention has advised the use of SARS-CoV-2 wastewater surveillance for assessing the prevalence and transmission of SARS-CoV-2 in communities [16]. Besides toxic substances and pathogenic microorganisms, hospital wastewater contains 2–3 times the amount of chemical and biological pollutants as urban wastewater, especially chemical demand of oxygen (COD), biological demand of oxygen (BOD), and suspended solids (SS) [17]. In addition, hospital wastewater contains a diverse range of micropollutants and macropollutants [17]. Micropollutants included hormones, detergents, antiseptics, antibiotics (such as sulfamethoxazole, ciprofloxacin, and paracetamol), absorbable organic halogens (AOX), contrast substances, phenols, analgesics, cytostatic, as well as heavy metals (such as iron, zinc, cadmium, chrome, copper, nickel, and lead). Macropollutants have physical-chemical parameters such as pH, total demand of oxygen (TOC), biological demand of oxygen, chemical demand of oxygen, ammonium ions and chloride, suspended solids, microbiological contaminants as coliforms, bacteria (enterococcus, shigella, salmonella), and viruses, especially SARS-CoV-2. This wastewater is discharged from various hospital units such as operating rooms, laboratories, laundries, kitchens, patient rooms, and research labs [17,18]. Recent studies have stated that SARS-CoV-2 should be monitored in community wastewater and hospital wastewater to help prevent COVID-19 outbreaks [3,5,19,20]. Therefore, wastewater surveillance for SARS-CoV-2 was implemented.

The identification of SARS-CoV-2 genetic targets in numerous kinds of samples using the molecular technique of reverse transcriptase quantitative polymerase chain reaction (RT-qPCR) is the current benchmark for COVID-19 diagnosis [21]. We presented a method for detecting SARS-CoV-2 in wastewater during the initial stages of the pandemic, when the prevalence of reported COVID-19 cases in Thailand was negligible [22]. Although this method is sensitive, the use of RT-qPCR to detect SARS-CoV-2 in wastewater requires specialized laboratory equipment and qualified staff and can take time. Several sensitive and user-friendly techniques for detecting SARS-CoV-2 have been investigated. Among them, chemiluminescence enzyme immunoassay (CLEIA) was developed which is fully automated and sensitive for the identification of SARS-CoV-2 N-Protein using a unique two-reaction. The sample and sample treatment solution are incorporated with an anti-SARS-CoV-2 monoclonal antibody-coated magnetic particle solution and incubated for 10 min at 37 °C to enable the formation of particular antigen–antibody immunocomplexes. An alkaline phosphatase-labelled anti-SARS-CoV-2 monoclonal antibody solution is included to the second reaction (available after washing) and incubated at 37 °C for 10 min in order to permit selective binding to the antigen of the aforementioned immunocomplexes and the production of subsequent immunocomplexes. Lastly, a substrate solution is mixed in and incubated for 5 min at 37 °C prior to the chemiluminescence signals being automatically read by the instrument and utilized in determining the amount of SARS-CoV-2 antigen in the sample using interpolation with a SARS-CoV-2 Ag calibrator curve [23]. Based on a recent study, CLEIA has the ability to detect and measure SARS-CoV-2 nucleocapsid protein from both nasopharyngeal swabs and saliva samples [23–26]. CLEIA is widely used for COVID-19 screening in Japanese airports, and it has been used for community and population screening to identify SARS-CoV-2 nucleocapsid protein in specimens (nasopharyngeal swabs) [27]. Furthermore, we established the potential role of CLEIA for identifying SARS-CoV-2 antigen in fresh market wastewater [6].

The goal of this work was finding SARS-CoV-2 in hospital wastewater by implementing RT-qPCR and CLEIA assay. The antigen results of CLEIA were then compared to the results of an RT-qPCR test targeting SARS-CoV-2 genomic RNA to evaluate these assays.

## 2. Materials and Methods

The study protocols were implemented in accordance with World Health Organization biosafety guidelines [28]. Likewise, the research project received authorization by Mahidol University's Institution Biosafety Committee (MU 2021-002).

(1)  Sample collection

SARS-CoV-2 wastewater surveillance was implemented for two buildings, the Administration Building and the Research and Welfare Building, in Ramathibodi Hospital from February to December 2022. Weekly grab samples of wastewater were obtained in sanitized bottles from each building, delivered to the laboratory on ice, and maintained at 4 °C until further investigation.

(2)  Sample preparation and concentration

To remove particles, a subsample (100–400 mL) of each collected wastewater grab sample was centrifuged at $3000 \times g$ for 10 min at room temperature. The supernatant was then filtered through a mixed cellulose ester membrane filter (pore size, 0.45 µm; diameter, 47 mm; GE Healthcare, Chicago, IL, USA) attached to a disposable Millicup™-FLEX filtration unit (Merck Ltd., Darmstadt, Germany). A vacuum pump was connected to the assembly filtration to filter the sample. Then, membrane filter was removed and set in a sterile 5 mL tube. Each sample tube received 1 mL of DNA/RNA Shield™ and 0.1 g of ZR BashingBeads (Zymo Research, Sigma, Irvine, CA, USA) prior to being maintained at −80 °C until further investigations.

(3)  SARS-CoV-2 identification and quantification via RT-qPCR

SARS-CoV-2 RNA extraction and RT-qPCR were performed to detect the ORF1ab, spike (S), and nucleocapsid (N) regions, as described in our previous research [22].

To elute the viral RNA, the prepared solution was first mixed 10 times (60 s each) with a vortex mixer at near-maximum speed. After that, the solution was then transferred in 400 µL to a fresh nuclease-free tube. The viral RNA was then extracted using the Viral RNA Mini Kit (Qiagen, Hilden, Germany) allowing compliance with the manufacturer's protocol. The sample was loaded onto the QIAamp Mini spin column which RNA bind to the membrane, whereas impurities were removed using two separate wash buffers (AW1 and AW2) in two brief centrifugations. The purified RNA free of protein, nucleases, other contaminants, and inhibitors was eluted in a special RNase-free buffer, ready for direct use or safe storage. A Nanodrop™ (Thermo Fisher Scientific, Waltham, MA, USA) was used to determine the purity and concentration of the extracted RNA. The nucleic acid purity was assessed using absorbance values at 260 and 280 nm (260/280 ratio). The general range of the 260/280 ratio was 1.9–2.1.

TaqMan™ 2019 nCoV Assay Kit v1 (Thermo Fisher Scientific) was used for RT-qPCR. Each 25 µL RT-PCR reaction mixture included 6.25 µL of 4×TaqPath™ 1-Step RT-qPCR Master Mix, 1.25 µL of COVID-19 Real-Time PCR Assay Multiplex Solution, 12.5 µL of nuclease-free water, and 5 µL of extracted RNA. The RT-qPCR experiment was performed on a ViiA 7 Real-Time PCR apparatus (Applied Biosystems, Waltham, MA, USA) with the following thermocycling conditions: 2 min at 25 °C for UNG incubation to remove amplicon carryover, 15 min at 50 °C for reverse transcription, 2 min at 95 °C for predenaturation, and 40 cycles of 3 s at 95 °C and 30 s at 60 °C for denaturation, annealing, and extension. As a positive control and internal positive control, the TaqPath COVID-19 control (Thermo Fisher Scientific) and MS2 phage control were utilized, respectively. A negative control was DNase/RNase-free water. The results were considered positive for SARS-CoV-2 detection if the cycle threshold (Ct) value for two or more SARS-CoV-2 target genes was less than 37.

(4)    Detection of SARS-CoV-2 antigen

In parallel with detection and quantification by RT-qPCR, 100 μL of concentrated samples was analyzed for the antigen quantification. The Lumipulse G1200 automated immunoassay analyzer (Fujirebio) was used to assess a specific chemiluminescence-based immunoassay technique [23]. The results were categorized as positive for SARS-CoV-2 detection by applying the manufacturer's advised antigen concentration cutoff for nasopharyngeal samples as 1.34 pg/mL.

(5)    Statistical analysis

Sensitivity and specificity values were calculated using the RT-qPCR results as the reference and with the exclusion of inconclusive samples determined by CLEIA assay. Cohen's kappa (κ) coefficient of results between the two tests with 95% confidence intervals (CI) was calculated.

## 3. Results

The numbers of reported cases and deaths are shown in Supplementary Figure S1. The number of in-hospital cases and deaths with confirmed SARS-CoV-2 detection during this period, compared to Thailand and Bangkok province, increased in early February 2022 and peaked in early May 2022. The number of reported cases then decreased to less than 100 by mid-July. According to Thailand's relaxation of COVID-19 restrictions on 1 October 2022 and current regulation of hospitals, the numbers of confirmed cases are not being disclosed [29].

Wastewater samples collected from February to December 2022 from the Administration Building and the Research and Welfare Building at Ramathibodi Hospital were tested for SARS-CoV-2 by RT-qPCR and CLEIA (Table 1). A total of 92 wastewater grab samples were tested. In total, 24/92 (26.09%) of the samples tested positive for SARS-CoV-2 by RT-qPCR, defined as a Ct value of less than 37 for two or more SARS-CoV-2 target genes. By contrast, 39/92 (42.39%) of the samples tested positive in the SARS-CoV-2 CLEIA antigen test, defined according to the manufacturer's antigen cutoff concentration (1.34 pg/mL) for nasopharyngeal swab samples.

As shown in Table 2, the RT-PCR results for 92 wastewater grab samples are compared to those produced by the CLEIA assay. Taking RT-qPCR as the reference assay, CLEIA returned 13 true-positive, 26 false-positive, 11 false-negative, and 42 true-negative results. Thus, the overall agreement rate of CLEIA was 59.8% (95% CI: 49.8–69.8%) with a sensitivity of 54.2% (95% CI: 44.0–64.3%) and specificity of 61.8% (95% CI: 51.8–71.7%). The κ coefficient = 0.1325 indicated slight agreement between CLEIA and RT-qPCR.

**Table 1.** SARS-CoV-2 detections in wastewater grab samples from Ramathibodi Hospital from February to December 2022.

| Date Collected | Administration Building | | | | | | Research and Welfare Building | | | | | |
| --- | --- | --- | --- | --- | --- | --- | --- | --- | --- | --- | --- | --- |
| | RT-qPCR (Ct Value) | | | | CLEIA | | RT-qPCR (Ct Value) | | | | CLEIA | |
| | N | ORF1ab | S | Interpretation | Antigen Concentration (pg/mL) | Interpretation | N | ORF1ab | S | Interpretation | Antigen Concentration (pg/mL) | Interpretation |
| 15-Feb-2022 | UD | UD | UD | neg | 11.73 | pos | UD | UD | UD | neg | 0.65 | neg |
| 22-Feb-2022 | 31.77 | 33.4 | 33.34 | pos | 0.42 | neg | UD | UD | UD | neg | 1.53 | pos |
| 1-Mar-2022 | 33.73 | 35.01 | 37.11 | pos | 0.42 | neg | 37.25 | UD | UD | neg | 0.56 | neg |
| 8-Mar-2022 | UD | 36.5 | 37.99 | neg | 0.39 | neg | UD | UD | UD | neg | 0.41 | neg |
| 15-Mar-2022 | UD | 36.54 | 34.86 | pos | 0.48 | neg | UD | UD | UD | neg | 0.06 | neg |
| 22-Mar-2022 | UD | 35.59 | UD | neg | 0.35 | neg | UD | 38 | UD | neg | 0.57 | neg |
| 29-Mar-2022 | 34.09 | 33.6 | 32.96 | pos | 3.2 | pos | 29.12 | 30.31 | 28.89 | pos | 2.14 | pos |
| 5-Apr-2022 | 38.36 | 38 | 33.96 | neg | 0.38 | neg | 31.08 | 32.49 | 31.49 | pos | 0.58 | neg |
| 11-Apr-2022 | UD | UD | 34.4 | neg | 2.38 | pos | 33.76 | 33.69 | 27.22 | pos | 5.79 | pos |
| 19-Apr-2022 | UD | 34.28 | 29.33 | pos | 2.34 | pos | UD | 35.32 | UD | neg | 0.22 | neg |
| 26-Apr-2022 | 30.34 | 32.99 | 32.05 | pos | 32.86 | pos | UD | UD | UD | neg | 15.09 | pos |
| 3-May-2022 | UD | 38 | UD | neg | 0.17 | neg | UD | UD | 37.69 | neg | 1.13 | neg |
| 10-May-2022 | UD | UD | UD | neg | 1.15 | neg | UD | UD | UD | neg | 0.86 | neg |
| 17-May-2022 | UD | 34.69 | 34.27 | pos | 0.69 | neg | UD | 38 | UD | neg | 0.53 | neg |
| 24-May-2022 | 31.42 | 33.28 | 33.24 | pos | 0.61 | neg | UD | UD | UD | neg | 1.28 | neg |
| 31-May-2022 | 34.22 | 38 | 35.89 | pos | 0.94 | neg | UD | UD | UD | neg | 1.1 | neg |
| 7-Jun-2022 | 37.63 | 33.452 | UD | neg | 1.05 | neg | UD | UD | UD | neg | 1.55 | pos |
| 14-Jun-2022 | UD | 37.96 | UD | neg | 0.7 | neg | UD | UD | UD | neg | 1.09 | neg |
| 21-Jun-2022 | UD | 35.93 | UD | neg | 1.93 | pos | UD | UD | UD | neg | 1.11 | neg |
| 28-Jun-2022 | 34.34 | 32.85 | 32.79 | pos | 0.46 | neg | UD | UD | 33.97 | neg | 2.21 | pos |

**Table 1.** *Cont*.

| Date Collected | Administration Building | | | | | | Research and Welfare Building | | | | | |
|---|---|---|---|---|---|---|---|---|---|---|---|---|
| | RT-qPCR (Ct Value) | | | | CLEIA | | RT-qPCR (Ct Value) | | | | CLEIA | |
| | N | ORF1ab | S | Interpretation | Antigen Concentration (pg/mL) | Interpretation | N | ORF1ab | S | Interpretation | Antigen Concentration (pg/mL) | Interpretation |
| 5-Jul-2022 | 34.11 | 34.38 | 28.73 | pos | 1.04 | neg | UD | UD | UD | neg | 2.99 | pos |
| 12-Jul-2022 | UD | 34.99 | UD | neg | 2.7 | pos | UD | UD | UD | neg | 2.73 | pos |
| 19-Jul-2022 | 32.19 | 35.16 | UD | pos | 2.66 | pos | 34.72 | UD | UD | neg | 1.13 | neg |
| 26-Jul-2022 | UD | UD | UD | neg | 1.21 | neg | UD | 36.28 | UD | neg | 0.65 | neg |
| 2-Aug-2022 | 35.23 | 35.48 | UD | pos | 0.42 | neg | 36.86 | UD | UD | neg | 0.82 | neg |
| 9-Aug-2022 | UD | UD | UD | neg | 0.88 | neg | UD | UD | UD | neg | 1 | neg |
| 16-Aug-2022 | UD | UD | UD | neg | 0.82 | neg | UD | UD | UD | neg | 1.45 | pos |
| 23-Aug-2022 | UD | 37.98 | UD | neg | 0.7 | neg | UD | UD | UD | neg | 0.71 | neg |
| 30-Aug-2022 | UD | UD | UD | neg | 0.3 | neg | UD | UD | 34.18 | neg | 0.39 | neg |
| 6-Sep-2022 | UD | UD | UD | neg | 0.07 | neg | UD | UD | UD | neg | 6.01 | pos |
| 13-Sep-2022 | UD | 35.73 | 38.11 | neg | 1.10 | neg | UD | UD | UD | neg | 4.82 | pos |
| 19-Sep-2022 | 39.94 | 39.85 | UD | neg | 1.15 | neg | UD | UD | UD | neg | 4.05 | pos |
| 26-Sep-2022 | UD | 37.97 | UD | neg | 2.8 | pos | UD | UD | UD | neg | 1.48 | pos |
| 3-Oct-2022 | 39.63 | 39.38 | UD | neg | 1.41 | pos | UD | UD | UD | neg | 2 | pos |
| 11-Oct-2022 | 33.31 | UD | UD | neg | 0.75 | neg | 34.29 | UD | UD | neg | 4.46 | pos |
| 18-Oct-2022 | 31.99 | 34.7 | UD | pos | 4.82 | pos | UD | UD | UD | neg | 3.9 | pos |
| 25-Oct-2022 | 31.98 | 34.96 | 32.89 | pos | 3.85 | pos | 33.99 | UD | UD | neg | 3.94 | pos |
| 1-Nov-2022 | 32.9 | UD | 33.82 | pos | 1.99 | pos | UD | UD | UD | neg | 1.22 | neg |
| 8-Nov-2022 | 36.52 | 39.35 | UD | neg | 1.58 | pos | 24.78 | 24.22 | UD | pos | 3.02 | pos |
| 15-Nov-2022 | 33.14 | 35.32 | UD | pos | 1.15 | neg | UD | UD | UD | neg | 2.52 | pos |

**Table 1.** *Cont.*

| Date Collected | Administration Building | | | | | | Research and Welfare Building | | | | | |
| | RT-qPCR (Ct Value) | | | | CLEIA | | RT-qPCR (Ct Value) | | | | CLEIA | |
| | N | ORF1ab | S | Interpretation | Antigen Concentration (pg/mL) | Interpretation | N | ORF1ab | S | Interpretation | Antigen Concentration (pg/mL) | Interpretation |
|---|---|---|---|---|---|---|---|---|---|---|---|---|
| 22-Nov-2022 | UD | UD | UD | neg | 3.29 | pos | 34.79 | 35.01 | UD | pos | 2.14 | pos |
| 29-Nov-2022 | 37.1 | 35.25 | 36.76 | pos | 2.65 | pos | UD | UD | UD | neg | 1.29 | neg |
| 6-Dec-2022 | UD | 35.19 | UD | neg | 2.21 | pos | UD | UD | UD | neg | 1.3 | neg |
| 13-Dec-2022 | 36.83 | 35.65 | UD | pos | 2.33 | pos | UD | UD | UD | neg | 0.38 | neg |
| 20-Dec-2022 | UD | UD | UD | neg | 1.15 | neg | UD | UD | UD | neg | 0.89 | neg |
| 27-Dec-2022 | UD | UD | UD | neg | 0.96 | neg | UD | UD | UD | neg | 1.67 | pos |

Notes: CLEIA, chemiluminescence enzyme immunoassay; Ct, cycle threshold; RT-qPCR, reverse transcriptase quantitative polymerase chain reaction; UD, undetermined. Positive results were defined as antigen cutoffs greater than 1.34 pg/mL by CLEIA and a Ct value < 37 for two or more SARS-CoV-2 target genes by RT-qPCR.

**Table 2.** Comparison of the CLEIA antigen test and RT-qPCR results for SARS-CoV-2 detection in wastewater grab samples.

|  | RT-qPCR+ | RT-qPCR- | Total |
|---|---|---|---|
| CLEIA + (≥1.34 pg/mL) | 13 | 26 | 39 |
| CLEIA − (<1.34 pg/mL) | 11 | 42 | 53 |
| Total | 24 | 68 | 92 |

## 4. Discussion

Wastewater surveillance is crucial for detecting and evaluating pathogens and viruses, especially SARS-CoV-2. The use of wastewater as a community surveillance tool has been widely recognized and accepted because it is non-intrusive and cost-effective. Infectious SARS-CoV-2 has not been isolated from either raw or processed wastewater effluents. High decay rates in raw sewage, resulting in low amounts of live virus particles, will make detection challenging using typical filtration-elution processes. SARS-CoV-2 survival has been studied in filtered and unfiltered raw wastewater, as well as secondary effluent at room temperature, in order to better understand the persistence of SARS-CoV-2 and developing variations in wastewater. The time required to inactivate 90% of SARS-CoV-2 was 10.4, 10.8, and 18.3 h for unfiltered raw, filtered raw, and secondary effluent, respectively. Following first order kinetics, there was a steady decrease in virus infectivity in different wastewater matrices [30].

We previously reported that RT-qPCR [22] and CLEIA [6] could be used to detect SARS-CoV-2 in wastewater. In the present study, we determined the presence of SARS-CoV-2 in hospital wastewater samples using RT-qPCR to identify SAR-CoV-2 viral RNA and CLEIA to identify SAR-CoV-2 viral antigen.

The association between increased vaccination coverage and alterations in dominant variations of concern was explored. There will be no vaccine-derived shedding from the current commercial supply of COVID-19 vaccinations, which may complicate the interpretation of SARS-CoV-2 wastewater-based surveillance data. Additionally, emerging infections from people who have been immunized contribute significantly to wastewater signals and ought to be assessed in accordance with the evolving patterns of shedding from novel varieties of concern [31].

The collection method for wastewater samples plays a crucial factor in the detection of the SARS-CoV-2 RNA virus. In our study, we employed a specific point-in-time approach, using single grab samples for COVID-19 wastewater surveillance. This method was chosen due to its rapidity, efficiency, cost-effectiveness, and safety, and it does not require automated equipment. In other words, the study attempts to conduct the simple and non-invasive surveillance for detection of SARS-CoV-2. This study focused on observing the discharge of COVID-19 virus antigen in wastewater representing at a specific time, rather than continuously monitoring throughout the day. If there is a need for consistent monitoring at different times throughout the day, a more reliable method, such as composite sampling, for the collection of hospital wastewater samples should be utilized. Previous studies have shown that the collection of grab samples, as opposed to composite samples, provides reliable identification of similarities and sufficient information regarding the presence of the SARS-CoV-2 virus in wastewater samples [32,33]. Furthermore, a recent study has suggested that performing grab sampling between 8 a.m. and 10 a.m. exhibits less variability in viral RNA concentrations [34].

PCR-based wastewater surveillance of SARS-CoV-2 RNA does not have the same biases as clinical surveillance and can be used as a tool to monitor changes in a region's overall SARS-CoV-2 prevalence [20,35]. However, it has unique limitations/biases of its own. Thus, working in tandem as complementary sources of data can help to provide a comprehensive view of disease transmission in communities. The wastewater-based surveillance's determining limitation was being unable to track virus lineage prevalence or establish epidemiological transmission links, which require understanding of the genome

sequence [36]. The number of COVID-19 cases needed in a population to generate a detectable viral RNA signal in wastewater was indicated. Li et al. (2023) gathered and analyzed a unique and big dataset composed of a significant number of diagnostic tests for COVID-19 and wastewater analyses spanning the first three waves of COVID-19 in Alberta, Canada. Results showed that a minimum of 4–17 (median 8), 9–43 (median 18), and 17–97 (median 38) daily reported new COVID-19 cases per 100,000 population were requirements for SARS-CoV-2 RNA to be found in community wastewater with 50%, 80%, and 99% probability, respectively [37]. In a previous study, we demonstrated the performance of RT-qPCR to detect three different SARS-CoV-2 genes (ORF1ab, the N protein gene, and the S protein gene) in wastewater from a market and a hospital even in high ambient temperature and relatively low prevalence of COVID-19 [22]. In the current study, we also applied this method to detect SARS-CoV-2 and confirmed that the virus can be detected in hospital wastewater. However, the interpreted positive results were not significantly correlated with newly confirmed COVID-19 cases on each water sampling date. It is possible that chemicals that are used in the hospital impair the detection efficiency. Zhang et al. reported that negative SARS-CoV-2 RNA results could be attributed to increased sodium hypochlorite supplementation used for disinfection [38]. Other disinfection methods, such as applying chlorine, chlorine dioxide, ozone, or UV irradiation, might also impair the detection efficiency [17]. Furthermore, the total volume of daily wastewater generated in buildings is not entirely correlated with the amount of people employing the wastewater system. Similar to our results, Tiacharoen et al. quantified SARS-CoV-2 RNA concentrations in hospital wastewater using a commercial clinical kit and RT-qPCR detection and found no significant correlation between SARS-CoV-2 RNA concentrations and newly confirmed COVID-19 cases on each wastewater collection date, pointing to a small number of SARS-CoV-2-positive samples [39]. Peng et al. demonstrate the high correlation between the viral signal of SARS-CoV-2 in wastewater with a time-varying relationship. In addition, the 15-day time lag between the average of SARS-CoV N1 and N2 gene concentrations and COVID-19 hospitalizations is adjusted for vaccination efforts [40]. The systematic review identified correlation between SARS-CoV-2 RNA concentration in wastewater and clinically confirmed cases, as demonstrated by Li et al. Among the 133 correlation coefficients, the systematic study report ranges from −0.38 to 0.99. Indeed, SARS-CoV-2 RNA concentration and new cases (whether daily new, weekly new, or future cases) had a stronger correlation than active cases and cumulative cases. Furthermore, environmental and epidemiological circumstances, as well as the wastewater-based epidemiology sampling strategy, could have an impact on these association coefficients. Wider changes in air temperature and clinical testing coverage, as well as an increase in catchment size, all had a significant negative impact on the connection between SARS-CoV-2 RNA concentration and COVID-19 case numbers [41].

The analytical capabilities of the CLEIA assay were determined in a previous investigation [6]. SARS-CoV-2 was identified in wastewater by the CLEIA assay downward to the $10^4$ spike cell, whereas RT-qPCR detected SARS-CoV-2 in wastewater downward to the $10^2$ spike cell. In addition, we used CLEIA to examine SARS-CoV-2 in 14 fresh market wastewater grab samples. The assay detected SARS-CoV-2 in the samples during an outbreak with 66.7% specificity and 100% sensitivity [6]. Therefore, in the present study, we used CLEIA to test for SARS-CoV-2 antigen in hospital wastewater. Using the manufacturer's recommended antigen threshold concentration for nasopharyngeal samples (1.34 pg/mL), we classified our data as positive for SARS-CoV-2 detection. The CLEIA results were compared with the RT-qPCR results as a standard control. SARS-CoV-2 was detected by CLEIA in 39/92 (42.39%) of samples, achieving a sensitivity and specificity of 54.2% (95% CI: 44.0–64.3%) and 61.8% (95% CI: 51.8–71.7%), respectively. Whenever we changed the cut-off level, the sensitivity and specificity did not improve. The newly identified SARS-CoV-2 genome mutations and a number of important variations, including multiple variants of concern (VOC), may influence to sensitivity and specificity of CLEIA. However, many reports were confirmed that the protein region detected by the CLEIA

assay is not altered by mutations. Roy established that the alpha and delta versions are distinguished by spike protein mutations; hence, CLEIA efficiency was unaffected [42]. Similar with Osterman et al., the CLEIA assay has applications for identifying alpha (B.1.1.7) or beta (B.1.351) [43]. Whereas CLEIA has been shown by Gandolfo et al. who succeeded to figure out the variant type and demonstrate amino acid substitutions within or near the functional N antigenic epitope, such as B.1.351, B.1.258, B.1.177.75, B.1.1.420, B.1.1.34, B.1.1.7, and P.1 [44]. The World Health Organization named omicron variants in November 2021, which contain mutations in spike proteins as well, and these variants will have no impact on the CLEIA results [45]. There was no statistically significant difference between the results of strains with mutations and SARS-CoV-2 antigen quantities [46,47].

The low specificity may have been owing to nonspecific obstructions from substances in the hospital wastewater, which might be alleviated by dilution [48], or it may have been affected by environmental factors such as temperature, pH, or the presence of chemical pollution [38]. CLEIA employs monoclonal antibodies directed against the SARS-CoV-2 N protein and is designed to perform double-antibody immunodiagnostic tests (sandwich assay) for the detection and quantification of SARS-CoV-2 antigen. However, the disinfectant substances used in the hospital may have significantly impacted the assay. The findings in this report are subject to at least four limitations. First, wastewater monitoring cannot offer a comprehensive view of disease transmission. Second, outpatient department interpretation of the results is constrained. Third, weekly grab wastewater cannot be representative of all due to hospitals generating large volumes of wastewater. Finally, pretreatment of wastewater surveillance can affect test results.

Recently, the growth and advancement of the digital revolution, including artificial intelligence, evolutionary computational, data science, big data, quantum science, bioinformatics, nanotechnology, internet of things (IoT), financial technology, and blockchain, create opportunities in the delivery of goods and services with higher revenue and a greater opportunity to combat the COVID-19 crisis. Predictive analytics based on data approaches (statistical analysis, machine learning, deep learning, and predictive models and algorithms) are a critical component of COVID-19 pandemic prediction and decision-making tools. Moreover, biosensors and smart sanitation technology are being developed to prevent, identify, and monitor new pollutants with global health potential at the communal (disease surveillance) and personal (for diagnostics) levels [49].

## 5. Conclusions

The presence of SARS-CoV-2 in hospital wastewater was confirmed by our findings. We found that 24/92 (26.09%) of hospital wastewater samples were positive for at least two SARS-CoV-2 genes (the ORF1ab, N, or S genes). However, unexpectedly, CLEIA had a low sensitivity and specificity of 54.2% (95% CI: 44.0–64.3%) and 61.8% (95% CI: 51.8–71.7%), respectively, as compared to RT-qPCR. The κ coefficient indicated slight agreement between assays. Based on the findings of our study, the CLEIA assay cannot replace molecular-based testing like RT PCR to determine SARS-CoV-2 in hospital wastewater, suspectedly due to hospital disinfectant substances. However, wastewater surveillance is and will continue to be a helpful tool for monitoring SARS-CoV-2 and providing an early warning indication. The application of computational modeling methods and big data analytics should be involved for the effective management of COVID-19.

**Supplementary Materials:** The following supporting information can be downloaded at: https://www.mdpi.com/article/10.3390/w15132495/s1, Figure S1: Number of reported cases and deaths. References [50–52] are cited in the supplementary materials.

**Author Contributions:** S.T., S.C. and S.P. contributed to samples collection and preparation, performed experiments, and wrote the manuscript. S.P., E.P. and B.O. contributed to designing the study concept, supervision, and critically revising the manuscript. All authors have read and agreed to the published version of the manuscript.

**Funding:** This research project is supported by Faculty of Medicine Ramathibodi Hospital Mahidol University (RF 65134).

**Data Availability Statement:** The data that support the findings of this study are available from the corresponding author upon reasonable request.

**Conflicts of Interest:** The authors declare that they have no conflicts of interest to disclose.

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
