# Peer review of "SARS-CoV-2 Surveillance in Hospital Wastewater: CLEIA vs. RT-qPCR"

_water, doi:10.3390/w15132495_

Round 1

Reviewer 1 Report

The authors presented an interesting study comparing RT-qPCR results with the CLEIA assay.  While not comprehensive in scope, the data presented would be useful to other individuals that may be considering using CLEIA in wastewater monitoring efforts.  The recommendation against using the CLEIA assay as statement by the authors is of value.  However, there are several points that need to be addressed in order to increase the clarity of the study.  They are listed below.

The title would benefit from being more descriptive by including CLEIA.  The use of CLEIA is the most novel aspect of this work and should be more prominently featured in the title.

Line 41:  While SARS-CoV-2 has been found on many surfaces, I am not aware of any studies demonstrating that infection is common, or even possible, from these surfaces.  I do not believe that the study cited makes this definitive conclusion either.  Thus, this sentence should be modified or removed.

Line 52:  SARS-CoV-2 cannot be spread through water or wastewater.  This has been definitively proven by many scientists across the globe.  This sentence should be deleted.

Lines 54/55:  References needed.

Line 70:  Again, studies have consistently found that SARS-CoV-2 in wastewater is not infectious.

Line 93: “from both and saliva” appears to be a typo.  Please correct.

Line 105:  What volume was collected during the grab samples?

Line 114:  How much supernatant was filtered?

Line 143:  What were the RT-qPCR cycling conditions?

Line 145:  Can the authors please elaborate on what was the “concentrated samples” used for the CLEIA assay?  It is difficult to tell if the CLEIA assay was more sensitive because it is accessing samples that have undergone greater concentrations steps than the RT-qPCR samples or if they are, indeed the exact same samples.

Line 172:  What is the rationale for requiring 2 or more target genes for calling RT-qPCR results positive?  Generally, only a single target is examined.  How do the results change if only one target is required to call a RT-qPCR target positive?  Similarly, what is the justification of using Ct values of 37 as the cut-off?  Many studies have successfully used higher cut-offs of around 39-40 Ct values.  Does this change the interpretation of your results if a higher Ct value was used?

Line 193:  Syntax needs to be corrected.

Line 200-203:  This is not a major transmission pathway of SARS-CoV-2 (if at all)

Line 210:  Syntax error.  Please correct.

Lines 210-222:  I’m not sure how this fits in with the rest of the study.  Were these contaminants quantified in the wastewater and was their impact on the RT-qPCR and CLEIA results assessed?

Line 226: Syntax error.  Please correct.

Line 234-236:  Where is the correlation data shown?  How was it conducted?

Line 236:  In addition to chemicals, the authors should point out that they used grab sampling which has been shown to be extremely variable when used at the building level in comparison with 24 hour composites.  Thus, the poor correlation could also be due to the high and low SARS-CoV-2 concentrations that come from grab samples.

Line 248:  What was this previous investigation?  Citation needed.

Line 259:  What were the new cutoff levels?

Reviewer 2 Report

The topic of this work is significant, but it seems to lack innovation and sufficient evidence to prove CLEIA had a low sensitivity and specificity compared to RT-qPCR. Overall, results were not sufficiently discussed and most of them were just simply described. Moreover, the manuscript would need an extensive editing of English language and a style to make it more comprehensive. Spelling mistakes and grammatical mistakes should be avoided. Besides, analyzing for obtained data is not enough. Parallel samples taken from the same day and same locations are required to generate a credible conclusion.

Specific comments:

1. Line space is not uniform. (e.g. lines 26-37 vs lines 38-42)

2. The data showed in Figure 1 is not generated by this paper but public. Only simple description is improper. Moving this part to supplementary file is better.

3. Detection of SARS-CoV2 listed in Table 2 should be given more analysis instead of just showing the result.

4. Discussion in lines 199-209 is not very close to this article. Please delete them.

5. Lines 210-212 might be better in introduction.

The manuscript would need an extensive editing of English language and a style to make it more comprehensive. Spelling mistakes and grammatical mistakes should be avoided.

Reviewer 3 Report

The authors present an original research article describing a comparison of two assays to detect SARS-CoV-2 in wastewater matrices. The content may be interesting to readers and the audience of the journal, however, there are several concerns that should be addressed prior to publication.

Abstract:

-       There appears to be a disconnect between the title of the study, which implies SARS-CoV-2 surveillance in a hospital, and what is presented here in the abstract, which seems to have a focus on comparing between RT-qPCR and CLEIA. Please make these two confluent.

Introduction:

-       Way too much extra background information that is unnecessary. It is well established that wastewater is a tool used to support public health decision making in monitoring for SARS-CoV-2

-       Line 51: It is difficult to tell what the authors mean by “Coronavirus transmission into water and wastewater” – please rephrase

o   Please explain how SARS-CoV-2 that attaches to masks of COVID-19 patients ends up in water and wastewater

-       I suggest to completely remove lines 1-50 in the introduction. Begin by acknowledging this work has been done by several groups throughout the world, list the current gaps in knowledge, then move on to introduce the purpose of the presented study

-       Line 54: Remove the word “Recently” from this sentence. It has been over three years since detecting and monitoring SARS-CoV-2 in wastewater has been “established”.

-       Please cite relevant sources after the sentence in lines 61-64 where the early warning potential of wastewater surveillance has been established

-       Please elaborate on what these “toxic substances” are mentioned in lines 68-71. Give specific examples in order to make this statement relevant.

-       I would like to see more of an explanation as to why a hospital was specifically chosen for this study

Methods

-       Can you please elaborate on what you mean by “weekly grab samples”? Were these samples collected once on the same day/time each week, or some other cadence?

-       Please explain why grab samples were chosen rather than 24-hour composite samples

-       Please list the standard curve used for RT-qPCR. What is the y-intercept?

-       Describe how LOD was established for the RT-qPCR assay

Results

-       It would be great if the authors would report within the text of the results section the numbers of reported cases and deaths that correspond with Figure 1

-       Please expand on the results within the text throughout the entire section

-       Table 2 – I think it would also be good to have a column that compares the sensitivity and specificity of each assay

Discussion

-       Line 185: Please revise – pathogens include viruses so they should not be listed separately here

-       Line 210: the word “micropollutants” is listed twice

-       Line 226: Please remove the word “And” from the beginning of the sentence

-       Line 224: Indeed, wastewater-based surveillance does not have the same biases as clinical surveillance, however, it has unique limitations/biases of its own. Thus, working in tandem as complementary sources of data can help to provide a comprehensive view of disease transmission in communities

-       Remove “infectious” from the sentence in line 227. A pathogen is generally an agent that can cause disease and thus this is redundant.

-       I don’t see any obvious or exclusively reported limitations from this study. Please add.

Conclusions

-       Instead of restating the specificity and sensitivity results which has already been done in the results and discussion, I’d like to see the authors draw a bit more out of the conclusions here. What might they suggest in future work? 

Suggest the authors revise the manuscript in its entirety for English editing.

Round 2

Reviewer 2 Report

no additional comments

Reviewer 3 Report

The authors have succeeded in addressing the previous concerns. I have no further edits or comments to provide.